# ALE: ADAPTIVE LENGTH EMBEDDING FOR TEXT RETRIEVAL

## ABSTRACT

Dense retrieval has become the dominant paradigm in modern text retrieval. It involves encoding text passages into high-dimensional vectors using an embedding model, and computing similarity via the dot product between query and passage vectors. For semantically complex texts, higher-dimensional vectors are often required to adequately capture their meaning. However, increasing vector dimensionality leads to higher storage costs and greater computational burden during online retrieval, which impacts its applicability in resource-constrained environments. In this paper, we analyze the embeddings generated by mainstream dense retrieval models and observe that they tend to exhibit significant redundancy, with high correlations among vector dimensions. To address this issue, we propose ALE, an Adaptive-Length Embedding method that produces variable-length vector representations tailored to the semantic complexity of each individual text. Specifically, ALE applies a linear transformation to convert the original embeddings into representations with linearly independent dimensions. It then selects the minimal number of dimensions necessary to preserve the semantic content of each text. To compute similarity between variable-length vectors, ALE adopts a hybrid approach by dividing each vector into a dense part and a sparse part. We conduct experiments on four datasets and demonstrate that ALE can reduce the average vector length by 75% and retrieval time by 84.8%, with minimal loss in retrieval performance. Furthermore, compared to the best dense baseline models with the same vector dimensionality ($d$=768), ALE achieves an improvement of 20.5% in nDCG@10.

## 1 INTRODUCTION

Text retrieval entails the process of searching for and obtaining relevant textual documents from an extensive collection based on a specified query. It is essential in numerous applications, such as search engines, recommendation systems, and question-answering systems. Recently, with the advent of LLMs, text retrieval has emerged as a pivotal element in retrieval-augmented generation (RAG) systems. The efficacy of text retrieval techniques hinges on the representation of textual data and the criteria used to assess the relevance between a query and the documents.

Dense retrieval has emerged as an effective approach in text retrieval due to its ability to capture high-level semantic features. It typically consists of two stages: encoding and interaction. During the encoding stage, the semantic features of the text are represented as dense vectors. In the interaction stage, the semantic similarity between the query and document vectors is computed. To facilitate the pre-building of indexes and to enhance online retrieval using approximate nearest neighbor (ANN) algorithms, most existing methods Karpukhin et al. (2020); Zhang et al. (2024); Neelakantan et al. (2022) employ a highly simplified interaction mechanism, such as the dot product. This design implies that the quality of vector encoding is crucial for the overall retrieval performance.

Similar to the scaling laws for LLMs Kaplan et al. (2020), the performance of dense embedding models is intricately linked to both the number of model parameters and the dimensionality of the encoded vectors Fang et al. (2024). For semantically complex documents or tasks that involve intricate definitions of similarity, high-dimensional representations are essential to adequately capture necessary semantic information Luan et al. (2021). However, the use of higher-dimensional vectors results in increased storage costs and extended query times during online retrieval, which adversely affects their applicability in resource-constrained environments, such as edge and end-site devices.

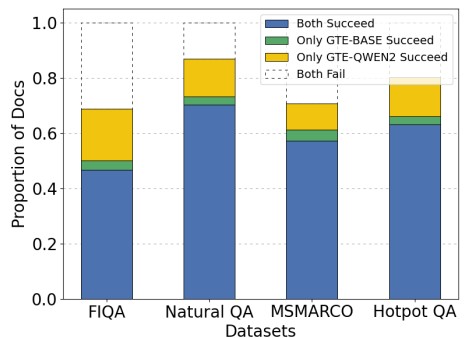

Figure 1: Proportion of documents successfully retrieved by both models, only by GTE-QWEN2, only by GTE-BASE, or by neither, across four datasets.

Common ANN algorithms, such as LSH Jafari et al. (2021) and IVFPQ Jégou et al. (2011b), generally exhibit query time complexity that scales linearly with the vector dimensionality $d$. In large-scale document retrieval scenarios, this renders the computational and storage overhead of high-dimensional vectors increasingly prohibitive.

Through data analysis, we observed that the high-dimensional dense embeddings generated by current models exhibit significant information redundancy. We compared the queries correctly retrieved by both high- and low-dimensional embedding models. Specifically, the high-dimensional model we used is the 3584-dimensional GTE-QWEN2-INSTRUCT Li et al. (2023), while the low-dimensional model is the 768-dimensional GTE-BASE. As illustrated in Figure 1, both models successfully retrieved relevant documents in most cases (i.e., the relevant document appears in the top 10 of the retrieval list). Only a small subset of semantically complex queries showed improved results with high-dimensional embeddings. This indicates that low-dimensional vectors are generally sufficient for capturing the semantic content of most documents, while only a limited number of semantically complex documents require high-dimensional representations. Nevertheless, current embedding models adopt fixed-length encodings and cannot dynamically adjust the embedding dimension according to the semantic complexity of individual documents.

To address the limitations of fixed-length dense embeddings, we propose ALE, an Adaptive-Length Embedding method. The core concept of ALE is to leverage the redundancy present in high-dimensional encoded vectors by transforming them into linearly independent components. For each document embedding, only the minimal number of dimensions necessary to retain its semantic information are preserved, ensuring that the dimensionality correspond to its semantic complexity. ALE operates on vectors encoded by existing high-dimensional embedding models, transforming them into representations with a significantly reduced dimensionality. As a result, it achieves storage and retrieval costs comparable to those of low-dimensional embeddings, while maintaining retrieval accuracy similar to that of high-dimensional embeddings.

ALE consists of two main steps: learning a transformation matrix and performing hybrid encoding. In the first step, ALE fits a transformation matrix using the original high-dimensional embeddings, ensuring that the transformed embeddings are linearly independent across dimensions while preserving the similarity between any two text embeddings before and after transformation. During the hybrid encoding phase, ALE divides each document embedding into a shorter dense part and a variable-length sparse part. By adjusting the length of the sparse part for different documents, the embedding length can be adapted to the semantic complexity of each document. During retrieval, ALE computes similarities by combining the dense and sparse components.

We evaluate ALE on four text retrieval datasets: FIQA2018 Maia et al. (2018), Natural QA Kwiatkowski et al. (2019), MSMARCO Bajaj et al. (2018), and HotpotQA Yang et al. (2018). We apply ALE to high-dimensional embeddings generated by SOTA LLM-based models, specifically the 3584-dimensional GTE-QWEN2-INSTRUCT and the 4096-dimensional BGE-EN-ICL. ALE transforms these high-dimensional vectors into variable-length embeddings. We compare their performance with fixed-dimensional embeddings obtained from dense models in the BGE and GTE series. The results show that ALE can reduce the average vector length and retrieval time by 75% and

84.8% respectively compared to the original high-dimensional embeddings, while preserving retrieval accuracy. With average dimensions of 768 and 1024, ALE achieves mean nDCG@10 scores that are 20.5% and 17.5% higher than those of the corresponding dense baseline models. This demonstrates that ALE improves storage and retrieval efficiency without sacrificing accuracy.

The primary contributions of this research are as follows:

- We propose an adaptive embedding method that converts fixed-length dense vectors into variable-length representations, adapting to each document's semantic complexity.
- We design a hybrid encoding method for adaptive-length embeddings to improve storage and retrieval efficiency.
- We conducted experiments to evaluate ALE on four text retrieval datasets, showcasing its superiority over existing fixed-length dense retrieval approaches.

## 2 RELATED WORK

Current text retrieval methods can be broadly categorized into term-based models, dense retrieval models, and LLM-based models.

### 2.1 TERM-BASED MODELS

Term-based models encode each document as a sparse vector over the vocabulary, where each dimension reflects statistical features such as term frequency in the document. The most common approaches in this category are TF-IDF and its improved variant BM25 Robertson and Zaragoza (2009). Due to their effectiveness and efficiency, these methods have been widely adopted in search engines. However, term-based models are limited to exact keyword matching. They cannot handle synonyms or capture semantic meaning at the sentence or paragraph level. As a result, their performance is suboptimal when retrieval relies on deeper semantic understanding.

### 2.2 DENSE RETRIEVAL MODELS

Dense retrieval models leverage deep neural networks, particularly pretrained language models (e.g. BERT Devlin et al. (2019)), to generate continuous vector representations of queries and documents. Unlike traditional term-based retrieval models that rely on keyword matching, dense retrieval enables semantic retrieval by mapping queries and documents into a shared dense embedding space.

As a foundational work in dense retrieval, DPR Karpukhin et al. (2020) introduces the dual-encoder model, which pre-encodes all documents to build an index and utilizes ANN algorithms Johnson et al. (2017); Liu et al. (2015); Jégou et al. (2011a;b) to expedite retrieval. This facilitates real-time access to collections containing millions of documents. Following DPR, several advanced dense retrieval models have emerged. ANCE Xiong et al. (2020) enhances performance through hard negative mining, treating top-ranked negative samples from ANN as hard negatives to improve contrastive learning. This refinement minimizes the discrepancy between data distributions during training and inference. RocketQA Qu et al. (2021); Ren et al. (2021) employs knowledge distillation, using a cross-encoder model as a teacher to generate soft labels for training the dual-encoder. These methods have proven effective and are widely adopted in training dense retrieval models.

BGE Xiao et al. (2024) and GTE Zhang et al. (2024) are two open-source retrieval model series that are continuously updated and incorporate the aforementioned training techniques. According to the MTEB leaderboards mte (2025); Muennighoff et al. (2022), these models attain the highest ndcg@10 scores among models with comparable parameter sizes and embedding lengths. Therefore, we choose them as baseline models for comparison in our study.

### 2.3 LLM-BASED MODELS

Recent advances in LLMs have led to dense retrieval models built on pre-trained LLMs Zhuang et al. (2024); Liu et al. (2024); BehnamGhader et al. (2024). Promptreps Zhuang et al. (2024) designs prompts requiring LLMs to summarize text with a single token, using hidden states and

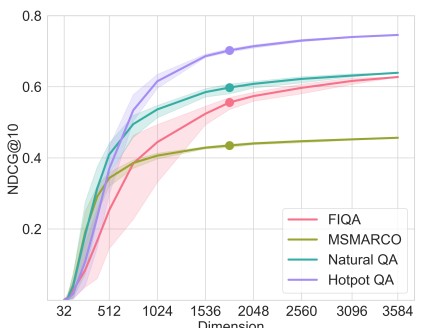

Figure 2: NDCG@10 score on the four datasets as a function of embedding dimensionality. High-dimensional embeddings from the GTE-QWEN2-INSTRUCT model are reduced via randomly selection; even with up to 50% dimensionality reduction (bold points), retrieval accuracy remains relatively stable. Shaded regions indicate performance range across 10 trials.

logit distributions as dense and sparse embeddings. Other models fine-tune pretrained LLMs under supervision, achieving state-of-the-art results. NV-EMBED Lee et al. (2025) improves representation by replacing standard attention with bidirectional attention. BGE-ICL Li et al. (2024) exploits in-context learning, providing examples to enhance embeddings. GTE-QWEN2 Zhang et al. (2024) fine-tunes Qwen2 and applies curriculum learning Xiong et al. (2024) to extend context length.

Although LLM-based models typically score higher than traditional dense models, their larger embedding dimensions can reduce efficiency in storage and online retrieval. ALE leverages the advantages of both model types by converting the high-dimensional embeddings from LLM-based models into variable-length vectors with lower average dimensions. This approach maintains nearly the same retrieval effectiveness while significantly decreasing storage costs.

## 3 METHODOLOGY

### 3.1 OVERVIEW

Dense embeddings often exhibit high correlation among different dimensions. On the four datasetsBajaj et al. (2018); Maia et al. (2018); Kwiatkowski et al. (2019); Yang et al. (2018), we randomly selected a subset of dimensions from the 3584-dimensional embeddings generated by the GTE-QWEN2-INSTRUCT model for retrieval. As shown in Figure 2, retrieval accuracy remains stable even when only half of the dimensions are used. This suggests that current high-dimensional embeddings can be further compressed. Additionally, as discussed in Section 1, increasing embedding dimensionality does not always enhance retrieval performance; the optimal dimensionality should align with the semantic complexity of documents. Mismatches between embedding dimensionality and semantic requirements can lead to decreased retrieval accuracy.

Based on these observations, we propose ALE, an **A**daptive-**L**ength **E**mbedding method for documents. The core idea is to decorrelate embedding dimensions and select the minimal subset of dimensions for each document required to sufficiently capture its semantic features. ALE consists of two steps: learning a transformation matrix and generating variable-length vectors. Given the original $d$-dimensional embeddings matrix $\boldsymbol{X} \in \mathbb{R}^{n \times d}$ of $n$ documents, we compute a transformation matrix $P$ such that the transformed matrix $\boldsymbol{X'} = \boldsymbol{X}\boldsymbol{P}$ has linearly independent dimensions while preserving the similarity structure of the original embeddings. For each transformed document embedding $\boldsymbol{x'}$, ALE then selects the smallest possible subset of dimensions such that the ratio of the sum of their squared values to the total sum of squared values of $\boldsymbol{x'}$ exceeds a predefined threshold $\theta$.

However, the aforementioned method encounters two challenges. First, directly fitting the transformation matrix using all document embeddings incurs significant computational and memory costs. Common datasets often contain millions of documents, and to ensure retrieval effectiveness, ALE operates on embeddings with high original dimensionality. Given that the complexity of solving the transformation matrix is $O(nd^2)$, when both the number of documents $n$ and the embedding

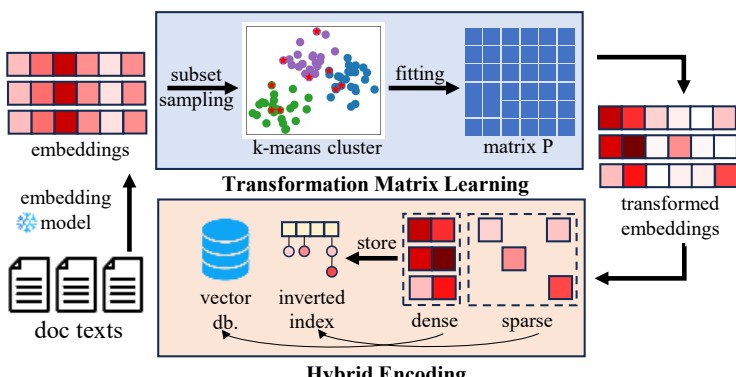

Figure 3: Overview of the ALE method

dimension $d$ are large, calculating the transformation matrix $\boldsymbol{P}$ becomes extremely time- and memory-intensive. Second, while adaptive-length embeddings preserve sufficient semantic features for each document, the varying number of dimensions retained across documents renders direct similarity computation unfeasible.

To address these challenges, we design a clustering-based sample strategy alongside a hybrid encoding scheme. First, for efficiency, ALE selects a subset of $s$ samples to estimate the transformation matrix $\boldsymbol{P}$. Specifically, we first perform k-means clustering on the document embeddings, then randomly select a sample subset from each cluster in proportion to its size, using all selected samples to fit $\boldsymbol{P}$. This approach ensures that, even with a limited number of samples, the chosen subset accurately reflects the overall dataset distribution. Second, after transforming the document embeddings, principal components that represent common and salient features are concentrated in the leading dimensions, while dimensions that represent unique characteristics of individual documents are assigned to later dimensions. Leveraging this property, ALE divides the transformed embeddings into a shorter, fixed-length dense part and a longer, variable-length sparse part, applying a hybrid similarity computation during retrieval. Figure 3 shows the framework of ALE.

## 3.2 Transformation Matrix Learning

In the first stage, we aim to transform the original document embeddings matrix $\boldsymbol{X} \in \mathbb{R}^{n \times d}$, where $n$ denotes the number of documents and $d$ represents the dimensionality of each embedding vector, into a new matrix $\boldsymbol{X'} \in \mathbb{R}^{n \times d}$ via a linear transformation $\boldsymbol{X'} = \boldsymbol{X}\boldsymbol{P}$. Here, $\boldsymbol{P} \in \mathbb{R}^{d \times d}$ is a square matrix to be solved.

The transformation $\boldsymbol{X'} = \boldsymbol{X}\boldsymbol{P}$ must meet two conditions. First, the dimensions should be independent. This means that the matrix $\boldsymbol{X'}^T \boldsymbol{X'}$ (i.e., the covariance matrix of the columns of $\boldsymbol{X'}$) must be diagonal, ensuring orthogonality among the transformed dimensions. Second, the similarity matrix after transformation should be same as the origin. Formally, the two constrains are formalized as:

$$\begin{cases} (\boldsymbol{X}\boldsymbol{P})^T(\boldsymbol{X}\boldsymbol{P}) = diag(\lambda_1, \lambda_2, ..., \lambda_d) \\ \|\boldsymbol{X}\boldsymbol{X}^T - \boldsymbol{X}\boldsymbol{P}\boldsymbol{P}^T\boldsymbol{X}^T\|_F = 0 \end{cases} \tag{1}$$

where $\|\cdot\|_F$ denotes the Frobenius norm.

We perform eigen-decomposition on the covariance matrix of the selected embeddings:

$$\boldsymbol{X}^T \boldsymbol{X} = \boldsymbol{V}\boldsymbol{\Lambda}\boldsymbol{V}^T \tag{2}$$

where $\boldsymbol{V} \in \mathbb{R}^{d \times d}$ is the orthogonal matrix of eigenvectors, and $\boldsymbol{\Lambda}$ is the diagonal matrix of eigenvalues. $\boldsymbol{P} = \boldsymbol{V}$ is a feasible solution that satisfies the two constraints in Eq. (1), and the transformed matrix $\boldsymbol{X'}$ is equivalent to applying PCA to $\boldsymbol{X}$, which means that the principal components are concentrated in the leading dimensions. We prove this transformation in appendix A.

To reduce the computational cost of deriving $\boldsymbol{P}$, we compute it using a subset of the original data. To ensure that this subset adequately represents the entire dataset, we adopt a clustering-based sampling strategy. Specifically, we first cluster the dataset $\boldsymbol{X}$ into $m$ clusters using K-means:

$$\{C_1, C_2, \ldots, C_m\}, \quad \bigcup_{i=1}^{m} C_i = \{\boldsymbol{x_1}, \ldots, \boldsymbol{x_n}\}. \tag{3}$$

Then, from each cluster $C_i$, we randomly sample a subset of data points proportional to the cluster size. Let $s$ be the total number of samples selected for computing $\boldsymbol{P}$, and let $|C_i|$ denote the number of samples in cluster $C_i$. Then the number of samples selected from $C_i$ is computed by:

$$s_i = \left\lfloor s \cdot \frac{|C_i|}{n} \right\rfloor \tag{4}$$

The selected samples from all clusters form the sampled matrix $\boldsymbol{X}_{\text{sub}} \in \mathbb{R}^{s \times d}$, which is then used to fit the transformation matrix $\boldsymbol{P}$.

## 3.3 HYBRID ENCODING

After applying the transformation matrix $\boldsymbol{P}$ to the original document embeddings, the resulting transformed embeddings have linearly independent dimensions. ALE aims to adapt the vector length of each document embedding to its semantic complexity by selecting the minimal number of dimensions that sufficiently capture its semantic features. To achieve this, ALE employs a straightforward strategy: it sets a predefined threshold $\theta$ and, for each transformed document embedding, applies a greedy algorithm to select dimensions. Specifically, the dimensions are first sorted in descending order of their absolute values, and then greedily selected from largest to smallest until the proportion of the cumulative squared sum of the selected dimensions to the total squared sum exceeds $\theta$.

However, the resulting variable-length vectors pose challenges for direct similarity computation. Different documents, due to varying semantic complexity, retain different numbers of significant principal components. Furthermore, because the transformation matrix is estimated from a subset of the dataset, the leading dimensions in the transformed space predominantly capture the global principal components across the data. Nonetheless, for a specific document embedding, significant values may still reside in later dimensions, reflecting document-specific semantic features. This causes the variable-length vectors of different documents to be misaligned, making direct dot product computation infeasible. While it is possible to store the indices and values of each document's selected dimensions in a sparse matrix format, the processes of compressing and reconstructing diminish the benefits of variable-length encoding.

To address this, we design a hybrid encoding method. Leveraging the property that principal components are predominantly located in the leading dimensions after transformation, we partition each transformed embedding into a dense part and a sparse part. For all document embeddings, the dense part is defined as the first $k$ dimensions of the transformed vector, where $k$ is a predefined hyperparameter. For the remaining dimensions, we apply the aforementioned greedy selection method to choose as few dimensions as possible for each document as the sparse part, ensuring that the combined sum of squares of the dense and sparse parts meets the predefined threshold.

Formally, for each vector $\boldsymbol{x'_i}$, we define the dense part as $\boldsymbol{x'_{i(dense)}} = (x'_{i,1}, \ldots, x'_{i,k})$. The sparse part is a variable-length vector, consisting of a subset of dimensions from the remaining dimensions $(x'_{i,k+1}, \ldots, x'_{i,d})$. The similarity between two document $\boldsymbol{x'_i}$ and $\boldsymbol{x'_j}$ is the sum of two components.

For the dense part, the similarity is the standard dot product over the first $k$ dimensions:

$$\text{sim}_{(dense)}(\boldsymbol{x'_i}, \boldsymbol{x'_j}) = \sum_{l=1}^{k} x'_{i,l} \cdot x'_{j,l}. \tag{5}$$

For the sparse part, the similarity is dot product over shared selected dimensions in the sparse part:

Table 1: Evaluation of nDCG@10 scores for each model.

| | MSMARCO | Natural QA | Hotpot QA | FIQA | Average |
|---|---|---|---|---|---|
| *768 dimension embeddings* | | | | | |
| BGE-768 | 41.35 | 54.16 | 72.59 | 40.64 | 52.18 |
| **ALE-BGE-768** | **45.99** | **67.96** | **78.69** | **58.79** | **62.86** |
| GTE-768 | 42.61 | 52.96 | 67.75 | 48.65 | 52.99 |
| **ALE-GTE-768** | **45.16** | **63.75** | **72.33** | **62.16** | **60.85** |
| *1024 dimension embeddings* | | | | | |
| BGE-1024 | 42.49 | 55.02 | 74.10 | 45.02 | 54.16 |
| **ALE-BGE-1024** | **46.21** | **68.82** | **80.33** | **59.22** | **63.64** |
| GTE-1024 | 42.93 | 56.08 | 68.18 | **63.22** | 57.60 |
| **ALE-GTE-1024** | **45.45** | **63.77** | **73.27** | 62.26 | **61.19** |
| *the original high-dimension embeddings* | | | | | |
| BGE-4096 | 46.48 | 70.02 | 82.06 | 59.79 | 64.59 |
| GTE-3584 | 45.66 | 63.95 | 74.59 | 62.75 | 61.74 |
| *ALE low-dimension embeddings* | | | | | |
| **ALE-BGE-512** | 45.07 | 65.04 | 78.10 | 57.51 | 61.43 |
| **ALE-GTE-512** | 44.27 | 63.13 | 70.61 | 61.60 | 59.90 |

$$\text{sim}_{(sparse)}(\boldsymbol{x'_i}, \boldsymbol{x'_j}) = \sum_{l \in S_i \cap S_j} x'_{i,l} \cdot x'_{j,l}. \tag{6}$$

where $S_i$ and $S_j$ are the selected dimension indices of $x'_i$ and $x'_j$, which is defined as:

$$S_i = \text{argmin}_{S_i \subseteq \{k+1,...,d\}} \left\{ \sum_{l \in S_i} (x'_{i,l})^2 \geq t \cdot \sum_{l=1}^{d} (x'_{i,l})^2 - \sum_{l=1}^{k} (x'_{i,l})^2 \right\} \tag{7}$$

Then, the total similarity is:

$$\text{sim}(\boldsymbol{x'_i}, \boldsymbol{x'_j}) = \text{sim}_{(dense)}(\boldsymbol{x'_i}, \boldsymbol{x'_j}) + \text{sim}_{(sparse)}(\boldsymbol{x'_i}, \boldsymbol{x'_j}). \tag{8}$$

To accelerate retrieval, we build an inverted index over the sparse part as shown in Figure 3, storing only the value of selected dimensions and their document IDs. During querying, it is only necessary to search the corresponding inverted index according to the dimensions selected in the sparse part of the query embedding. This enables fast sparse similarity computation without reconstructing full-length vectors, while the dense part can be computed efficiently using ANN algorithms.

## 4 EXPERIMENTS

### 4.1 EXPERIMENTAL SETUP

**Baseline Models and Datasets** In our experiments, we evaluated ALE using high-dimensional embeddings produced by SOTA models, and compared its performance with dense baselines. Specifically, we used embeddings from BGE-EN-ICL (4096 dims) and GTE-QWEN2-INSTRUCT (3584 dims) as originals, then compressed them into variable-length embeddings via ALE. The resulting ALE embeddings consist of a fixed-length dense component and a variable-length sparse component. In our evaluation, we varied the length of the dense component, conducting experiments with dimensions of 512, 768, and 1024 for both models. The sparsity threshold $\theta$ was set to 0.75 to achieve a balance between retrieval performance and accuracy. For dense baseline models, we selected four models from the BGE and GTE series with fixed embedding dimensions of 768 and 1024. The experiment utilized four datasets: FIQA2018 Maia et al. (2018), Natural QA Kwiatkowski et al. (2019), MSMARCO Bajaj et al. (2018), and HotpotQA Yang et al. (2018). The Details of baseline models, datasets and the implementation details are described in appendix B.

Table 2: Evaluation of recall@1000 scores for each model.

|  | MSMARCO | Natural QA | Hotpot QA | FIQA | Average |
|---|---|---|---|---|---|
| *768 dimension embeddings* | | | | | |
| BGE-768 | 97.55 | 98.58 | 94.23 | 90.82 | 95.29 |
| **ALE-BGE-768** | **99.21** | **99.78** | **95.84** | **97.90** | **98.18** |
| GTE-768 | 98.01 | 97.94 | 85.25 | 95.48 | 94.17 |
| **ALE-GTE-768** | **99.05** | **99.78** | **94.33** | **98.81** | **97.99** |
| *1024 dimension embddings* | | | | | |
| BGE-1024 | 97.80 | 98.94 | 95.48 | 92.51 | 96.18 |
| **ALE-BGE-1024** | **99.27** | **99.78** | **96.59** | **98.02** | **98.42** |
| GTE-1024 | 98.30 | 98.97 | 85.29 | 98.10 | 95.16 |
| **ALE-GTE-1024** | **99.11** | **99.78** | **94.52** | **98.78** | **98.05** |
| *the original high-dimension embeddings* | | | | | |
| BGE-4096 | 99.30 | 99.78 | 97.01 | 98.19 | 98.57 |
| GTE-3584 | 99.18 | 99.78 | 94.96 | 98.77 | 98.17 |
| *ALE low-dimension embeddings* | | | | | |
| **ALE-BGE-512** | 99.00 | 99.77 | 95.73 | 97.53 | 98.01 |
| **ALE-GTE-512** | 98.85 | 99.67 | 93.81 | 98.77 | 97.78 |

Table 3: Evaluation of time cost per query (ms) for each model.

|  | MSMARCO | Natural QA | Hotpot QA | Fiqa | Average |
|---|---|---|---|---|---|
| *768 dimension embeddings* | | | | | |
| dense-768 | 1371.8 | 405.7 | 720.7 | 9.2 | 626.8 |
| **ALE-768** | 1385.9 | 411.0 | 724.4 | 9.2 | 632.6 |
| *1024 dimension embeddings* | | | | | |
| dense-1024 | 1997.8 | 533.1 | 1060.1 | 10.7 | 900.4 |
| **ALE-1024** | 1997.9 | 533.1 | 1060.2 | 10.7 | 900.5 |
| dense-3584 | 9049.3 | 2346.2 | 5064.0 | 42.8 | 4125.6 |
| dense-4096 | 11329 | 2703.8 | 5715.3 | 48.7 | 4949.2 |
| **ALE-512** | **937.7** | **325.1** | **491.6** | **8.1** | **440.6** |

**Metrics** We evaluated retrieval accuracy using NDCG@10 and Recall@1000, which are widely used in recent literature and benchmarks such as MTEB Muennighoff et al. (2022). Firstly, NDCG@10 measures the ranking quality of the top 10 retrieved results, defined as:

$$\text{NDCG@10} = \frac{\text{DCG@10}}{\text{IDCG@10}} = \frac{1}{\text{IDCG@10}} \sum_{i=1}^{k} \frac{rel_i}{\log_2(i+1)} \tag{9}$$

where $rel_i$ is the relevance of the result at rank $i$, and IDCG@10 is the ideal DCG. Additionally, Recall@1000 is calculated as:

$$\text{Recall@k} = \frac{\text{Number of relevant documents in top } k}{\text{Total number of relevant documents}} \tag{10}$$

These metrics respectively reflect the precision of top-ranked results and the model's ability to retrieve all relevant documents. Retrieval cost was measured by the storage space (GB) required for embeddings and the latency per query (ms) during online retrieval.

## 4.2 EVALUATION RESULTS

Table 1 and Table 2 present the results for ndcg@10 and recall@1000, respectively. The six fixed-dimensional embeddings are represented using the format "series name-dimension", such as BGE-768. The hybrid embeddings generated by ALE are denoted as ALE-BGE-d and ALE-GTE-d, where d represents the dimension of the dense component and the embeddings are derived from BGE-4096 and GTE-3584 as input, respectively.

Table 4: Storage cost (GB) of all documents for each model.

|  | MSMARCO | Natural QA | Hotpot QA | Fiqa | Average |
|---|---|---|---|---|---|
| *768 dimension embeddings* | | | | | |
| dense-768 | 27.16 | 8.23 | 16.07 | 0.17 | 12.91 |
| **ALE-768** | 27.57 | 8.33 | 16.27 | 0.17 | 13.09 |
| *1024 dimension embeddings* | | | | | |
| dense-1024 | 36.21 | 10.98 | 21.43 | 0.23 | 17.21 |
| **ALE-1024** | 36.22 | 10.98 | 21.44 | 0.23 | 17.22 |
| dense-3584 | 126.76 | 38.44 | 75.02 | 0.82 | 60.26 |
| dense-4096 | 144.86 | 43.93 | 85.74 | 0.94 | 68.87 |
| **ALE-512** | **18.99** | **5.80** | **11.23** | **0.12** | **9.04** |

The experimental results demonstrate that ALE consistently produces hybrid representations that outperform fixed-length dense baselines of the same dimensionality. For instance, at 768 dimensions, ALE-BGE-768 and ALE-GTE-768 achieve average ndcg@10 improvements of 20.5% and 14.8% over BGE-768 and GTE-768, respectively. Similarly, at 1024 dimensions, ALE-BGE-1024 and ALE-GTE-1024 deliver improvements of 17.5% and 6.2% over BGE-1024 and GTE-1024, respectively.

Furthermore, the retrieval performance of hybrid embeddings obtained by ALE is highly correlated with the performance of the original high-dimensional embeddings on each dataset. For instance, due to differences in the training process, BGE-EN-ICL achieves stronger results on Natural QA and HotpotQA, while GTE-QWEN2-INSTRUCT performs better on FIQA. This trend is preserved after dimensionality reduction by ALE.

Table 3 and Table 4 show the time (ms) processing a single query during online retrieval and the storage space (GB) needed for all document embeddings. Since for dense models, time and space depend solely on the dimension and not on the specific model used, we omitted the model names. The experiments revealed that ALE consumes more time and space than dense models at the same dimension due to the additional resources required for storing and retrieving the sparse part. However, since the sparse part uses an inverted index and the sparse components of most text embeddings are short when the threshold $t$ is low, the additional time and space overhead is minimal. For the MSMARCO dataset at 768 dimensions, the increase in time and space consumption was only 1%. Moreover, as the dense dimension increases, the ratio of the squared sum of the dense part to the total squared sum also increases, further reducing the overhead of the sparse part. At a dense dimension of 1024, the additional time and space consumption for the sparse part becomes negligible.

Moreover, we explored the potential of further reducing the dimensionality of the dense component in ALE to verify whether it could outperform fixed-dimensional embeddings in both retrieval accuracy and efficiency. As shown in Tables 1 - 4, ALE-BGE-512 and ALE-GTE-512 not only require less retrieval time and storage space compared to dense-768 but also achieve higher retrieval accuracy than fixed-dimensional embeddings with 1024 dimensions. This demonstrates that, in contrast to fixed-length dense embeddings, ALE can enhance retrieval accuracy while simultaneously reducing the storage and retrieval costs of embeddings.

The ablation studies are included in Appendix C.

## 5 CONCLUSION

In this paper, we propose ALE, an Adaptive-Length Embedding method designed to address the inefficiencies of fixed-length embedding models by generating variable-length vectors tailored to the semantic complexity of individual texts. By transforming high-dimensional embeddings into linearly independent components and retaining only the necessary dimensions, ALE significantly reduces storage and computational costs. Experimental results on four benchmark datasets demonstrate that ALE achieves an average vector length reduction by 75%f compared to the original, with minimal performance loss. Moreover, ALE outperforms baseline dense embeddings with comparable dimensionality, achieving an 20.5% improvement in nDCG@10. These findings emphasize ALE's potential to improve the efficiency and scalability of contemporary text retrieval systems.

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

APPENDIX

## A  THEORETICAL PROOF

In Section 3, we introduced the transformation matrix $\boldsymbol{P} = \boldsymbol{V}$ via the spectral decomposition of $\boldsymbol{X}^T\boldsymbol{X}$. Here, we prove that $\boldsymbol{V}$ satisfies the necessary conditions for $\boldsymbol{P}$ by demonstrating its three essential properties. These properties validate ALE's efficacy in converting dense text embeddings into variable-length representations. We assume $\boldsymbol{X} \in \mathbb{R}^{n \times d}$ is centered, $\boldsymbol{X}^T\boldsymbol{X} = \boldsymbol{V}\boldsymbol{\Lambda}\boldsymbol{V}^T$, and $\boldsymbol{X} = \boldsymbol{U}\boldsymbol{\Sigma}\boldsymbol{V}^T$ is its single value decomposition (SVD).

**Theorem 1.** *After applying the transformation $\boldsymbol{V}$ to $\boldsymbol{X}$, the **covariance matrix** of the transformed embedding matrix $\boldsymbol{X}' = \boldsymbol{X}\boldsymbol{V}$ is diagonal, indicating that its dimensions are uncorrelated. That is, given $\boldsymbol{X}' = \boldsymbol{X}\boldsymbol{V}$, the covariance matrix is:*

$$Cov(\boldsymbol{X}') = \frac{1}{n-1}\boldsymbol{X}'^T\boldsymbol{X}' = \frac{1}{n-1}\boldsymbol{\Lambda}, \tag{11}$$

*where $\boldsymbol{\Lambda}$ is diagonal.*

*Proof.* First, we compute the Gram matrix of the transformed data:

$$\boldsymbol{X}'^T\boldsymbol{X}' = (\boldsymbol{X}\boldsymbol{V})^T(\boldsymbol{X}\boldsymbol{V}) = \boldsymbol{V}^T\boldsymbol{X}^T\boldsymbol{X}\boldsymbol{V}. \tag{12}$$

Then, substitute $\boldsymbol{X}^T\boldsymbol{X} = \boldsymbol{V}\boldsymbol{\Lambda}\boldsymbol{V}^T$:

$$\boldsymbol{X}'^T\boldsymbol{X}' = \boldsymbol{V}^T(\boldsymbol{V}\boldsymbol{\Lambda}\boldsymbol{V}^T)\boldsymbol{V} = (\boldsymbol{V}^T\boldsymbol{V})\boldsymbol{\Lambda}(\boldsymbol{V}^T\boldsymbol{V}) = \boldsymbol{\Lambda}, \tag{13}$$

since $\boldsymbol{V}^T\boldsymbol{V} = \boldsymbol{I_d}$. Since $X$ is centered, the sample covariance is:

$$\mathrm{Cov}(\boldsymbol{X}') = \frac{1}{n-1}\boldsymbol{X}'^T\boldsymbol{X}' = \frac{1}{n-1}\boldsymbol{\Lambda}, \tag{14}$$

a diagonal matrix with entries $\lambda_i/(n-1)$. Thus, the dimensions of $X'$ are uncorrelated. $\square$

**Theorem 2.** *The transformation $\boldsymbol{X}' = \boldsymbol{X}\boldsymbol{V}$ exactly preserves the original similarity matrix, as indicated by the Frobenius norm of the difference between $\boldsymbol{X}\boldsymbol{X}^T$ and $\boldsymbol{X}'\boldsymbol{X}'^T$ being zero. Formally, for $\boldsymbol{X}' = \boldsymbol{X}\boldsymbol{V}, \|\boldsymbol{X}\boldsymbol{X}^T - \boldsymbol{X}'\boldsymbol{X}'^T\|_F = 0$.*

*Proof.* From the SVD, $\boldsymbol{X} = \boldsymbol{U}\boldsymbol{\Sigma}\boldsymbol{V}^T$, so $\boldsymbol{X}' = \boldsymbol{X}\boldsymbol{V} = \boldsymbol{U}\boldsymbol{\Sigma}\boldsymbol{V}^T\boldsymbol{V} = \boldsymbol{U}\boldsymbol{\Sigma}$. The transformed similarity matrix is:

$$\boldsymbol{X}'\boldsymbol{X}'^T = (\boldsymbol{U}\boldsymbol{\Sigma})(\boldsymbol{U}\boldsymbol{\Sigma})^T = \boldsymbol{U}\boldsymbol{\Sigma}\boldsymbol{\Sigma}^T\boldsymbol{U}^T. \tag{15}$$

And the original similarity matrix is:

$$\boldsymbol{X}\boldsymbol{X}^T = (\boldsymbol{U}\boldsymbol{\Sigma}\boldsymbol{V}^T)(\boldsymbol{U}\boldsymbol{\Sigma}\boldsymbol{V}^T)^T = \boldsymbol{U}\boldsymbol{\Sigma}\boldsymbol{V}^T\boldsymbol{V}\boldsymbol{\Sigma}^T\boldsymbol{U}^T = \boldsymbol{U}\boldsymbol{\Sigma}\boldsymbol{\Sigma}^T\boldsymbol{U}^T. \tag{16}$$

Thus

$$\|\boldsymbol{X}\boldsymbol{X}^T - \boldsymbol{X}'\boldsymbol{X}'^T\|_F = \|\boldsymbol{0}\|_F = 0, \tag{17}$$

Which means that the similarity matrix is fully preserved. $\square$

**Theorem 3.** *For a reduced dimensionality $k < d$, the transformation using the first $k$ columns of $V$, denoted $V_k = V[:,:k]$, minimizes the similarity loss $\|XX^T - X'X'^T\|_F$ where $X' = XV_k$, concentrating information in the leading dimensions. For $X' = XV_k$:*

$$\|\boldsymbol{X}\boldsymbol{X}^T - \boldsymbol{X}'\boldsymbol{X}'^T\|_F = \min_{\boldsymbol{P} \in \mathbb{R}^{d \times k}} \|\boldsymbol{X}\boldsymbol{X}^T - (\boldsymbol{X}\boldsymbol{P})(\boldsymbol{X}\boldsymbol{P})^T\|_F. \tag{18}$$

Table 5: Configuration of the baseline models

| Model Name | Architecture | Dimension | Parameter Size | Max Input Tokens |
|---|---|---|---|---|
| BGE-BASE-EN-V1.5 | encoder-only | 768 | 109M | 512 |
| BGE-LARGE-EN-V1.5 | encoder-only | 1024 | 335M | 512 |
| BGE-EN-ICL | decoder-only | 4,096 | 7B | 32,768 |
| GTE-BASE-EN-V1.5 | encoder-only | 768 | 137M | 8,192 |
| GTE-LARGE-EN-V1.5 | encoder-only | 1,024 | 434M | 8,192 |
| GTE-QWEN2-INSTRUCT | decoder-only | 3,584 | 7B | 32,000 |

*Proof.* Let $\boldsymbol{X}' = \boldsymbol{X}\boldsymbol{V_k} = \boldsymbol{U}\boldsymbol{\Sigma}\boldsymbol{V}^T\boldsymbol{V_k}$. Since $\boldsymbol{V}^T\boldsymbol{V_k} = [I_k; 0]$:

$$\boldsymbol{X}' = \boldsymbol{U}\begin{bmatrix}\boldsymbol{\Sigma_k} \\ \boldsymbol{0}\end{bmatrix}, \tag{19}$$

where $\boldsymbol{\Sigma_k} = \mathrm{diag}(\sigma_1, \ldots, \sigma_k)$. Then:

$$\boldsymbol{X}'\boldsymbol{X}'^T = \boldsymbol{U}\begin{bmatrix}\boldsymbol{\Sigma_k^2} & \boldsymbol{0} \\ \boldsymbol{0} & \boldsymbol{0}\end{bmatrix}\boldsymbol{U}^T. \tag{20}$$

And the difference is:

$$\boldsymbol{X}\boldsymbol{X}^T - \boldsymbol{X}'\boldsymbol{X}'^T = \boldsymbol{U}\left(\begin{bmatrix}\Sigma^2 & 0 \\ 0 & 0\end{bmatrix} - \begin{bmatrix}\Sigma_k^2 & 0 \\ 0 & 0\end{bmatrix}\right)\boldsymbol{U}^T = \boldsymbol{U}\begin{bmatrix}0 & 0 \\ 0 & \Sigma_{k+1:r}^2\end{bmatrix}\boldsymbol{U}^T \tag{21}$$

where $\boldsymbol{\Sigma_{k+1:r}^2} = \mathrm{diag}(\sigma_{k+1}^2, \ldots, \sigma_r^2)$. The Frobenius norm is:

$$\|\boldsymbol{X}\boldsymbol{X}^T - \boldsymbol{X}'\boldsymbol{X}'^T\|_F = \sqrt{\sum_{j=k+1}^{r} \sigma_j^4}. \tag{22}$$

By the Eckart–Young–Mirsky theoremWikipedia contributors (2025), this is the minimal loss for a rank-$k$ approximation of $\boldsymbol{X}\boldsymbol{X}^T$, achieved by retaining the $k$ largest singular values.

Thus, $\boldsymbol{V}[:, :k]$ is optimal, concentrating information in the leading $k$ dimensions. □

## B  IMPLEMENTATION DETAILS

ALE was implemented in Python, with the open-source libraries $sentence - transformer$ and $faiss$ used for text embedding and dense vector retrieval, respectively. All base models used in the experiments were sourced from the open-source versions available on Hugging Face[1]. The configuration of baseline models are summarized in Table 5. The experimental environment was a Linux virtual machine with 240GB of memory and L20 GPUs. The GPUs were employed to accelerate the inference of text embeddings. The datasets are download from the MTEB space[2] of HuggingFace. For datasets with a provided training set (i.e., FIQA, MSMARCO, and HotpotQA), we used both the entire document corpus and the training queries to fit the transformation matrix. For datasets without a training set (i.e., Natural QA), only the document corpus was used.

To clarify the practical retrieval process, we outline the two-stage inference procedure in Algorithm 1. The process begins with an initial retrieval using the dense components to generate a candidate set of documents. This is followed by a re-ranking step that incorporates scores from the sparse components to produce the final ranked list. This hybrid approach ensures both efficiency and accuracy.

---

[1]https://huggingface.co/
[2]https://huggingface.co/mteb

**Algorithm 1** ALE Inference

**Require:** Query $q$, Document Collection $D$
**Ensure:** Top $k$ ranked documents
1: Encode $q$ into dense representation $q_{dense}$ and sparse representation $q_{sparse}$
2: **for** each document $d$ in $D$ **do**
3:     Compute dense score: $s_{dense}[d] \leftarrow$ dot_product$(q_{dense}, d_{dense})$
4: **end for**
5: Select top $2k$ documents $D'$ from $D$ based on $s_{dense}$
6: **for** each document $d$ in $D'$ **do**
7:     Compute sparse score: $s_{sparse}[d] \leftarrow$ dot_product$(q_{sparse}, d_{sparse})$
8:     Compute final score: $s_{final}[d] \leftarrow s_{dense}[d] + s_{sparse}[d]$
9: **end for**
10: Rerank documents in $D'$ by $s_{final}$ in descending order
11: **return** Top $k$ documents from $D'$

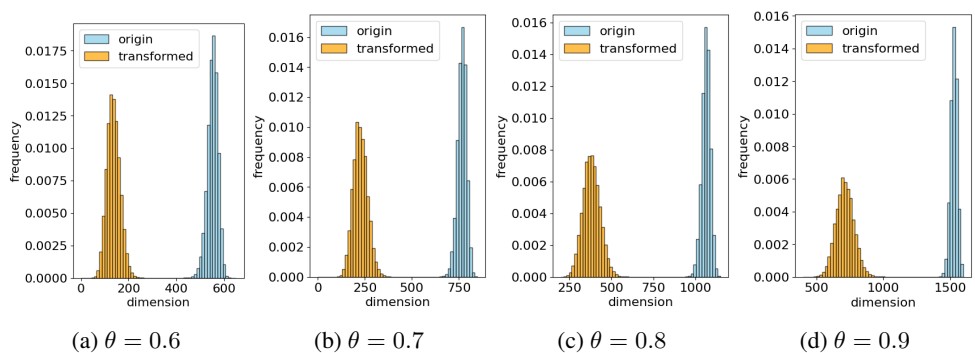

(a) $\theta = 0.6$    (b) $\theta = 0.7$    (c) $\theta = 0.8$    (d) $\theta = 0.9$

Figure 4: The number of dimensions required to achieve the threshold $\theta$ on the MSMARCO dataset, with and without the transformation.

## C    ABLATION ANALYSIS

### C.1    THE IMPACT OF EMBEDDING TRANSFORMATION

We also analyzed the impact of the transformation module on the distribution of embeddings. As illustrated in Figure 4, we calculated the minimum number of dimensions required to achieve the threshold for the squared sum ratio and plotted the frequency distribution histograms of the necessary dimension counts before and after transforming the original embeddings. The baseline model used was GTE-QWEN2-INSTRUCT, and the dataset was MSMARCO (we conducted experiments on other datasets, observing similar distribution patterns). The results demonstrate that the number of necessary dimensions decreased by over 50% after transformation. This indicates that the transformation module can concentrate the information distribution of embeddings more effectively.

### C.2    DENSE COMPONENT IN HYBRID ENCODING

Figure 5 presents the nDCG@10 scores for ALE retrieval when retaining different dimensionalities of the dense component. The points marked with circles indicate the data points corresponding to the highest nDCG@10 scores. As shown in Figure 5, reducing the dimensionality incurs almost no loss in retrieval accuracy when the original dimensionality is high. Results on the FIQA and Natural QA datasets even demonstrate a slight improvement in retrieval scores when the dimensionality of the dense component is slightly reduced from the original 3584 dimensions. This suggests that the variable-length vectors constructed by ALE are able to filter out noise present in the original encodings. The retrieval scores begin to drop sharply only when the dimensionality of the dense component falls below a certain threshold (around 1024 dimensions). This trend indicates that, when the dimensionality is above this threshold, the encoded vectors contain substantial redundant information, and dimensionality reduction causes minimal loss in representational capacity. However,

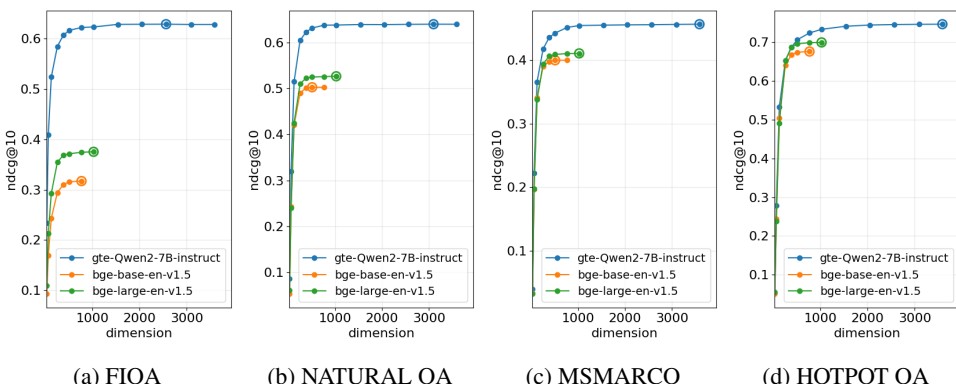

<p style="text-align:center">(a) FIQA      (b) NATURAL QA      (c) MSMARCO      (d) HOTPOT QA</p>

Figure 5: Ndcg@10 scores for all datasets across different dense dimensions.

when the dimensionality is reduced below the threshold, information loss becomes significant and retrieval accuracy declines markedly. Additionally, ALE was applied to two smaller models, BGE-BASE-EN-V1.5 and BGE-LARGE-EN-V1.5, with original dimensionalities of 768 and 1024, respectively. The results show that even for models with lower original dimensionality, there is still room for further dimensionality reduction. The critical dimensionality is consistent across all models, as are the retrieval scores observed below this threshold.

## C.3 COMPARISON WITH OTHER DIMENSIONALITY REDUCTION METHODS

To further contextualize ALE's performance, we compare it against three established supervised and unsupervised dimensionality reduction techniques.

We evaluated against PCA, Product Quantization (PQ), and a supervised Linear projection method, following the setup in Ma et al. (2021). We used BGE-4096 as the base model. For PCA, we reduced the dimensionality to 512. For PQ, we set the number of subquantizers $m = 256$ and bits per subquantizer $n_{bits} = 8$. For the Linear method, we trained a projection layer on labeled pairs for one epoch with a learning rate of $10^{-3}$.

The results in Table 6 show that ALE significantly outperforms these baselines. On average, ALE's nDCG@10 score is 14.25%, 12.74%, and 11.47% higher than the Linear, PQ, and PCA methods, respectively. It is worth noting that while the supervised Linear method performs well on large datasets, its performance degrades on smaller ones like FIQA and it requires costly labeled data. In contrast, ALE remains fully unsupervised while delivering superior accuracy.

Table 6: Comparison of nDCG@10 scores between ALE and other dimensionality reduction methods on BGE-4096 embeddings. ALE dimension is set to an average of 512.

| Method | MSMARCO | Natural QA | Hotpot QA | FIQA | Average |
|---|---|---|---|---|---|
| Linear-BGE-512 | 42.19 | 58.24 | 72.93 | 41.71 | 53.77 |
| PQ-BGE (m=256, n=8) | 40.77 | 56.05 | 71.23 | 49.90 | 54.49 |
| PCA-BGE-512 | 41.06 | 56.88 | 72.01 | 50.49 | 55.11 |
| **ALE-BGE-512** | **45.07** | **65.04** | **78.10** | **57.51** | **61.43** |

To provide a more direct comparison with the most related unsupervised method, PCA, we evaluated both methods at multiple target dimensions (512, 768, 1024). As shown in Table 7, ALE consistently achieves higher nDCG@10 scores across all tested dimensions, demonstrating its superior ability to preserve salient semantic information during compression.

## C.4 QUALITATIVE ANALYSIS OF ADAPTIVE LENGTHS

To provide an intuitive understanding of how ALE adapts embedding length to semantic complexity, we present four document examples from the MSMARCO dataset. We sorted all documents by

Table 7: Comparison of nDCG@10 scores between PCA and ALE at different target dimensions, using BGE-4096 as the base model.

| Method | MSMARCO | Natural QA | Hotpot QA | FIQA | Average |
|---|---|---|---|---|---|
| *PCA Method* | | | | | |
| PCA-BGE-512 | 41.06 | 56.88 | 72.01 | 50.49 | 55.11 |
| PCA-BGE-768 | 42.65 | 58.94 | 73.63 | 51.90 | 56.78 |
| PCA-BGE-1024 | 43.87 | 59.73 | 74.49 | 52.52 | 57.65 |
| *ALE Method* | | | | | |
| **ALE-BGE-512** | **45.07** | **65.04** | **78.10** | **57.51** | **61.43** |
| **ALE-BGE-768** | **45.99** | **67.96** | **78.69** | **58.79** | **62.86** |
| **ALE-BGE-1024** | **46.21** | **68.82** | **80.33** | **59.22** | **63.64** |

the number of non-zero dimensions in their sparse component (with a dense part of $k = 512$ and $\theta = 0.75$) and sampled one document from each quartile of the distribution.

As shown in Table 8, a correlation emerges between the number of required sparse dimensions and factors such as text length, the number of named entities, and the specificity of the content (i.e., general vs. domain-specific knowledge). Simpler, more general documents require fewer sparse dimensions, while longer, more complex documents with specialized terminology retain a larger number of dimensions to capture their unique semantic fingerprint.

Table 8: Examples of documents from the MSMARCO dataset with varying sparse components.

| Sparse Dimensions | Document Text |
|---|---|
| 19 | A group of tissues working together to perform a similar function is called an organ. Examples of organs found in the body are the skin, lungs, heart, kidneys and liver. |
| 39 | The name Davis is of English and English origin. The meaning of Davis is beloved. Davis is generally used as a boy's name. It consists of 5 letters and 2 syllables and is pronounced Da-vis. |
| 87 | Norman Jewison was well respected as a director, thanks to In the Heat of the Night (1967) and The Thomas Crown Affair (1968), two very tough and masculine films. Fiddler on the Roof was thus a formidable challenge as it was his first foray into the musical genre. Early on in the project, Jewison determined that the film's success depended on a strong element of realism throughout the production, hence his extensive travels from Canada to much of Eastern Europe. |
| 136 | Another leading industry there is government. Juneau became a state capital when Alaska became the 49th U.S. state in 1959, and today, nearly 60 percent of the city's population works in government. The governor's mansion stands on a hillside over-looking the cruise docks, and anyone can take a walk up the hills via steep stairways. The most exciting way to see the glacier is by helicopter. Temsco Helicopters (877-789-9501) offers a basic tour with about 30 minutes in the air and 20 to 25 minutes on the glacier; upgrade to the pilot's choice tour for two different glacier landings. |

