# OpenReview forum: "ALE: Adaptive Length Embedding for Text Retrieval"
_ICLR.cc/2026/Conference — ICLR 2026 Conference Withdrawn Submission_

### Official Review · Reviewer_ygH6 · 2025-10-31

**Soundness:** 2
**Presentation:** 2
**Contribution:** 2
**Rating:** 2
**Confidence:** 5

**Summary:**

The paper proposes length adaptation method to reduce the memory consumption of text embeddings by leveraging salient values within transformed embeddings. Experimental results suggest that the proposed approach can, to some extend, preserve the performance of advanced text embeddings such as BGE and GTE while improving storage efficiency.  However, there remain notable limitations in terms of technical novelty, adequacy of evaluation, and clarity of presentation.

**Strengths:**

- This paper proposes a length adaptation method designed to reduce the memory consumption of text embeddings.
- It addresses the redundancy issue inherent in popular high-dimensional embeddings by leveraging salient values within transformed embeddings for document-specific length adaptation.
- Experimental results demonstrate that the proposed method effectively preserves the performance of advanced text embeddings, such as BGE and GTE.

**Weaknesses:**

- Missing comparison with relevant studies. While there are well-established techniques for embedding dimension reduction—such as PQ, OPQ, AQ, DPQ, and JPQ—the paper does not compare its method against these baselines. The authors may refer to prior work such as *Jointly Optimizing Query Encoder and Product Quantization to Improve Retrieval Performance* for relevant methods.

- Misleading presentation of experimental results. BGE/GTE-768, -1024, and -4096 are distinct text embedding models, not the same model with varying reduced dimensions. However, the current presentation may easily lead readers to misinterpret them as embeddings derived from a single model with dimensionality reduction applied.

- Accuracy degradation. The proposed method results in a substantial drop in retrieval accuracy (e.g., NGCD@10). Given that the test corpus is relatively small, this accuracy loss is likely to be even more pronounced in large-scale retrieval scenarios.

- Limited practicality due to hybrid indexing. The reliance on a hybrid index structure poses integration challenges with existing search systems (e.g., ElasticSearch), thereby limiting the method’s applicability in real-world deployments.

**Questions:**

Please refer to the detailed comments provided above.

---

### Official Review · Reviewer_X1mT · 2025-11-01

**Soundness:** 2
**Presentation:** 2
**Contribution:** 2
**Rating:** 6
**Confidence:** 5

**Summary:**

The authors propose a combination of dimensionality reduction with adaptive-length vector encodings for text retrieval. Their technique, called ALE, leverages PCA and an inverted index based on the trailing PCA dimensions for reranking. The technique is simple in nature and yet appears effective.

**Strengths:**

- The finding that using a variable number of dimensions per vector is interesting.
- The method to select the number of dimensions is simple and yet appears effective.
- The inverted index over the trailing dimensions using the dominant trailing dimensions of the query is an intriguing solution over storing all trailing dimensions.

**Weaknesses:**

The description of the dimensionality reduction used in this work seems a bit inflated in my opinion. It is a well known fact that PCA/SVD linearly decorrelates data, but the authors seem to be re-deriving these properties for some reason. The proofs provided in Appendix A are very well known and learned in any basic linear algebra course. In my opinion, at this point, the properties of PCA/SVD should be given for granted and assumed as baseline knowledge for readers of machine learning papers.

Using a different number of dimensions per vector would create a deflation of the similarity between vectors (in the sense that such a projection decreases the distances between the vectors). It would be interesting to see an analysis of why having an uneven deflation/shrinkage does not present particular challenges over an even one. Since this is the key point of the paper, I would expect a deeper analysis around this point.

There are previous uses of dimensionality reduction for vector search that the authors should acknowledge and probably compare to. I leave some examples for reference below. I understand that ALE may offer footprint savings when compared to some of these approaches. However, the idea of doing a search in a low-dimensional space and a re-ranking in a higher-dimensional space is not new. The authors should compare their results against the existing techniques.

- Tepper, M., Bhati, I. S., Aguerrebere, C., Hildebrand, M., & Willke, T. L. LeanVec: Searching vectors faster by making them fit. Transactions on Machine Learning Research.
- Jegou, H., Douze, M., Schmid, C. and Perez, P. (2010), Aggregating local descriptors into a compact image
representation, in ‘IEEE Conference on Computer Vision and Pattern Recognition’, IEEE, pp. 3304–3311.
- Gong, Y., Lazebnik, S., Gordo, A. and Perronnin, F. (2012), ‘Iterative quantization: A procrustean approach to learning binary codes for large-scale image retrieval’, IEEE transactions on pattern analysis and machine intelligence 35(12), 2916–2929.
- Babenko, A. and Lempitsky, V. (2014b), ‘The inverted multi-index’, IEEE transactions on pattern analysis and machine intelligence 37(6), 1247–1260.
- Wei, B., Guan, T. and Yu, J. (2014), ‘Projected residual vector quantization for ANN search’, IEEE MultiMedia 21(3), 41–51.
- Zhang, H., Tang, B., Hu, W. and Wang, X. (2022), Connecting compression spaces with transformer for approximate nearest neighbor search, in ‘European Conference on Computer Vision’, pp. 515–530.

In practical deployments of RAG pipelines, re-indexing (or to incrementally update the index) is often needed. It would be interesting to see how the authors would handle such a case when the leading PCA dimensions can change over time.

I understand that the authors are not proposing a new index but a new vector representation. However, doing a lienar scan over the vectors is considered prohibitive in practice, even if fewer dimensions are used. It would be interesting to pair ALE with a standard index (and IVF or graph index) that operates on the low-dimensional space, and only activates the sparse component upon reranking. Such an experiment would provide a clearer idea of how ALE performs in a more realistic setting.

**Questions:**

Please refer to my remarks above.

---

### Official Review · Reviewer_GADZ · 2025-11-01

**Soundness:** 3
**Presentation:** 3
**Contribution:** 3
**Rating:** 4
**Confidence:** 3

**Summary:**

This paper propose ALE for adaptively reducing text embedding dimensionality. Given the orignial embeddings, ALE first applies a linear transformation to map them into embeddings with linearly independent dimensions. It then divides the dimensions into two parts: a dense part where the first-k dimensions are always kept, and a sparse part, where only a subset of dimensions remain active (non-zero) based on a threshold-based selection criterion. Extensive experiments across various models and datasets demonstrate the effectiveness of ALE.

**Strengths:**

(1) This paper focuses on the important problem of reducing text embedding diemensionality.

(2) The proposed approach of dividing the transformed dimensions into dense and sparse parts to improve efficiency while maintaining performance is interesting.

(3) Across various models and datasets, ALE demonstrates clear efficiency advantages.

(4) ALE also achieves notable improvements over the original embeddings for small models, demonstrating its advantages beyond efficiency.

**Weaknesses:**

(1) Since the paper focus on reducing dimensionality, Matryoshaka representation is relevant. However, it is not discussed in the paper. An experimental comparison may not be necessary, since Matryoshaka representation requires model training while ALE does not. But for ALE, there might be a demand for updating the transformation matrix when new embeddings are introduced. How could ALE deal with this scenario?

(2) I assume the retrieval experiments are conducted using the Flat index. However, ANN indices like HNSW can substantially improve efficiency while maintaining performance. Although ALE reduces dimensionality, it also introduces a two-stage retrieval pipeline. It would be interesting to see how a 4096-dimensional embedding with HNSW would compare to a dense-512-dimensional ALE embedding using the proposed two-stage setup, with HNSW also applied to the dense part.

(3) The comparisons with other diemsion reduction baseines in Tables 6 and 7 may not be entirely fair. Different from the baselines, ALE includes a sparse part that may contain additional information.

(4) The paper reports only the number of dimensions in the dense part, but does not provide information on how many dimensions in the sparse part are active on average. Moreover, there is no detail on the number of linearly independent dimensions produced by the transformation.

**Questions:**

(1) The motivation for dimension redundancy between L74 to L84 may not be entirely convincing, as the comparison involves two completely different models, introducing factors beyond dimensionality alone.

---

### Official Review · Reviewer_wvUk · 2025-11-01

**Soundness:** 1
**Presentation:** 2
**Contribution:** 2
**Rating:** 2
**Confidence:** 4

**Summary:**

The paper introduces Adaptive Length Embedding (ALE), a post-processing method to improve the efficiency of dense retrieval by converting fixed-length, high-dimensional vectors into variable-length representations based on each text's semantic complexity. ALE applies a linear transformation equivalent to PCA to de-correlate dimensions and then partitions each vector into a fixed-length dense part and an adaptively selected, variable-length sparse part. Experiments on four benchmark datasets show that ALE reduces average vector length and retrieval time while maintaining accuracy comparable to the original high-dimensional embeddings

**Strengths:**

1. The post-processing nature of the method makes it applicable on any given dense embedding model which makes the method widely applicable
2. Writing is overall easy to follow

**Weaknesses:**

### W1 - missing baselines and context
MRL [1] is a widely adopted technique to enable variable length embeddings in dense retrieval models, almost all embedding based apis (openai, google, etc) and recent dense embedding models employ this - but it is not mentioned or compared against.

### W2 - poor experimental setup
- (a) The main table compares lower d baselines which have a much weaker underlying transformer backbone with the proposed method - the key result "same vector dimensionality (d=768), ALE achieves an improvement of 20.5% in nDCG@10" is misleading, the d=768 model is a much smaller 109M parameter model while ALE's 768d model is derived from a 7B model, this is not a very valid setting to present the main results in my opinion.
- (b) The authors can correct me if I am wrong but as per my understanding the cost only represents the cost of doing nearest neighbor search - not the query encoding process, which is also not fair comparison.
- (c) No substantial analysis beyond embedding dimension comparison, missing ablation on hyperparameters such as $t$, effect of no dense part vs no sparse part, etc

### W3 - incremental proposed method
the idea of PCA on dense embeddings has been proposed extensively in literature, keeping a sparse component is an incremental addition which probably won't benefit from efficient dense vector search techniques or hardware (GPU/TPU)

### References
[1] Matryoshka Representation Learning, Kusupati et al

**Questions:**

1. For cost comparison, what does cost mean, is it just the cost of doing nearest neighbor search over the query vector and the document vectors (in this case what is the setup used for doing NNS?) or is the end-to-end search cost which also includes the cost of encoding a query? - for a 7B model I am assuming this cost is going to be significantly high and need to be considered
2. In line 50, the paper supports the statement "performance of dense embedding models is intricately linked to both the number of model parameters and the dimensionality of the encoded vectors" by citing Fang et al. (2024), to the best of my knowledge the scaling laws in Fang et al are for a fix sized (768d) embedding models - the paper doesn't show results for embedding dimension scaling right?

---

### Note · Authors · 2025-11-15

**Comment:**

We would like to withdraw our submission in order to further improve the design and experiments based on the reviewers’ valuable feedback. We sincerely appreciate the opportunity and thank the reviewers for their constructive comments.

**Withdrawal Confirmation:**

I have read and agree with the venue's withdrawal policy on behalf of myself and my co-authors.